# Congested Period in Professional Youth Soccer Players Showed a Different High Decelerations Profile in the Group Performance and a Specific Positional Behaviour

**DOI:** 10.3390/jfmk7040108

**Published:** 2022-11-29

**Authors:** Borja Muñoz-Castellanos, Alberto Rabano-Muñoz, Bernardo Requena, Luis Suarez-Arrones, Jose A. Asian-Clemente

**Affiliations:** 1Real Betis Balompié, Performance Department, 41012 Seville, Spain; 2Football Science Institute, 18016 Granada, Spain; 3Department of Sport Sciences, Universidad Pablo de Olavide, 41013 Seville, Spain; 4FC Lugano, Performance Departament, 6900 Lugano, Switzerland

**Keywords:** time-motion, running, fatigue, external load, GPS

## Abstract

Present soccer demands are increasing in terms of running requirements and the number of matches until youth soccer players experience several periods of fixture congestion during the season. Currently, congested periods have not been extensively studied in this population. For this reason, this study aimed to compare the running demands of professional youth soccer players in congested periods according to their specific playing positions. Twenty youth players were grouped according to their position: Central Defenders (CD), Fullbacks (FB), Midfielders (MF), Wide Midfielders (WM) and Strikers (ST). A GPS system was used to monitor the players during the first (M1), second (M2) and third (M3) matches played during a congested period, measuring their total distance covered (TDC), DC 18.0–20.9 km·h^−1^, DC 21.0–23.9 km·h^−1^, DC > 24.0 km·h^−1^, number of high accelerations (>2.5 m·s^−2^), number of high decelerations (<2.5 m·s^−2^) and peak speed (km·h^−1^). M1, M2 and M3 showed the same TDC, DC 18.0–20.9 km·h^−1^, DC 21.0–23.9 km·h^−1^, DC > 24.0 km·h^−1^, number of high accelerations, and peak speed (*p* > 0.05). The statistical analysis showed significant differences between M1, M2 and M3 in the decelerations recorded between M1 and M3 (*p* < 0.05). Likewise, each position showed specific behaviours during the congested period, with all showing at least one difference in DC 18.0–20.9 km·h^−1^, 21.0–23.9 km·h^−1^ or >24.0 km·h^−1^ between M1, M2 and M3 (*p* < 0.05). In conclusion, coaches should pay attention to the fatigue produced by the number of high decelerations. Secondly, an individualized training protocol should be considered according to the running requirements of each position when youth professional soccer players are involved in a congested period.

## 1. Introduction

Soccer is a highly intermittent sport where players’ activity is composed of high and low-intensity movements of varying lengths and situations depending upon an array of factors [1,2,3,4]. One of the most important aspects of modern elite soccer is the increasing demands in terms of running requirements and the number of matches played during the season [5]. The number of matches has increased in top-level European teams from around 50 matches in the 2008/2009 season to around 60 matches currently [6], meaning that professional players experience several periods of fixture congestion during the season [7]. A congested schedule is considered to exist when there is a minimum of two successive bouts of match-play, with an inter-match recovery period of less than 96 h [8]. Congested periods are common when soccer teams participate in more than one competition during the season or when national teams play in an international tournament [9,10,11].

Previous studies have shown that depending on the variable examined, professional soccer players may experience changes in their running requirements between matches during congested and non-congested periods. During congested periods, it has been affirmed that the distances covered at different running intensities in elite professional soccer remained unchanged between matches [12,13,14,15,16]. However, when congested periods are experienced, some researchers have found a difference in performance in terms of total distance, number of sprints, distance covered at different intensities, and acceleration and deceleration profiles [17,18,19]. Although congested periods are a common concern in elite professional soccer [20], they have not been extensively studied in the younger soccer population. In professional soccer academies, this type of schedule has recently received attention in specific age groups: Under 14, Under 15, Under 17 and Under 19 [21,22,23,24]. In these age groups, there were no differences between congested match periods and non-congested periods for the total distance and the distance covered >21 km·h^−1^ in the most demanding passages of the matches [23]. Variables including the numbers of accelerations and decelerations and mean metabolic power showed increases in congested periods compared to non-congested periods [22]. Another study indicated a decline in total distance covered and player load during the congested period [24]. Although these previous studies compared the differences between congested and non-congested periods, other investigations have analysed player behaviour during matches in the same congested periods. Some researchers found dissimilarities in the distance covered at different intensities, but no changes were found in sprint frequency [25], and the number of decelerations decreased in the last four matches played in a row in a congested period by elite youth soccer players [22]. In contrast, other studies showed that running performance at different intensities was maintained between matches 1 and 3 for under-23 soccer players [26]. Studies on this topic showed discrepancies in the results obtained. As some authors have found, these controversial outcomes may be due to a high variability between soccer matches [27,28,29]. This diversity in running performances between matches has been proven to be relatively “large” [30], reaching from 15% to >60% depending on the variable studied [31], thus highlighting the inherent context of soccer.

Similar congested periods occur when youth soccer players are evaluated in tournaments carried out over a short period of 3–4 days [22,32]. In a short tournament of this kind, there were no alterations found in the total distance covered, high-intensity running distance, or maximum running speed [8,22,33,34]. However, another study showed that accelerations were affected in this kind of short tournament [32].

Taking into account that the external load of soccer players is related to their playing position [27,28,35,36], the scientific literature has also evaluated running behaviour for different playing positions during congested periods. These investigations showed that there were differences between specific positions, especially in the total distance covered and the distance covered at moderate and high intensity [15,18,19,26,37].

To the authors’ knowledge, there is no research to date that has investigated the running performance of different playing positions for youth soccer players during a congested period. These periods are common in professional soccer, and they have received much attention in recent times due to an increase in matches in both national and international championships. In youth soccer players, these congested periods have not been studied extensively during a regular championship. This may be because these types of periods have not previously been experienced by young players. According to the rising interest in performance development, youth soccer players have been involved in systematic training programs and matches designed to progress player promotion [38]. In order to better understand the physical demands required to play professional soccer, it is necessary to delineate the match demands in this framework [39]. These types of investigations are needed to clarify running requirements in youth players during adult soccer situations. The aim of this study was thus to compare the external load of professional youth soccer players in matches played during a congested period and evaluate the positional running requirements for each match.

## 2. Materials and Methods

### 2.1. Subjects

Twenty male youth soccer players from one Spanish professional academy participated in this study (Age = 15.95 ± 1.85 years, Height = 175.6 ± 5.35 cm, Body Weight = 63.17 ± 6.9 kg). A total of 4 players were selected from the Under-14 team (mean ± SD, 13.25 ± 0.5 years; 172.53 ± 4.12 cm; 56.95 ± 4.29 kg), 7 from the Under-16 team (mean ± SD, 15.29 ± 0.49 years; 172.93 ± 4.55 cm; 59.01 ± 5.43 kg) and 9 from the Under-19 team (mean ± SD, 17.67 ± 0.71 years; 179.04 ± 4.66 cm; 69.18 ± 3.08 kg). They were grouped according to their positional roles as Central Defenders (CD, *n* = 6), Fullbacks (FB, *n* = 5), Midfielders (MF, *n* = 3), Wide Midfielders (WM, *n* = 3) and Strikers (ST, *n* = 3). All players, regardless of their team or position, participated in 5 soccer training sessions per week (strength and conditioning and technical-tactical sessions) and normally competed in a single weekly match except during the congested period where they played three matches in the same week. All players were declared injury-free and fit for competition by medical staff prior to participation in any match. These data were acquired as a condition of player monitoring, in which player activities are measured over the competitive season [40], so ethics committee clearance was not required. Nevertheless, the study conformed to the recommendations of the Declaration of Helsinki, and the participants were informed of the study’s design and aims, giving their consent before it started.

### 2.2. Design

The running demands of 21 official matches (Under 14 = 3, Under 16 = 9 and Under 19 = 9) during the 2020–2021 season were monitored, resulting in 60 player match observations. These observations were subsequently partitioned into 3 different types of matches according to their temporal distribution and were categorized as M1 for the first matches of congested periods, M2 for the second and M3 for the third. Only 7-day microcycles were included in this study, excluding those where the matches were disputed with a duration of more than 72 h with each other. Players were only included in the analysis if they played for ≥75% of the total match duration [22,32]. Those who did not fulfil this criterion were excluded from the data analysis. All assessed matches were part of the Regular Championship for each team, and there were different match durations for each age category: 2 halves of 40 min with a 15-min half-time interval for those Under 14, and 2 halves of 45 min with a 15-min half-time interval for those Under 16 and those Under 19. All matches were played on artificial grass and under similar environmental conditions. Substitutions were also different for each age category: for the Under-14 group, rotative substitutions were permitted, while coaches allowed the Under-16 and Under-19 groups a maximum of 5 substitutions, respectively. None of the teams used systematic post-match recovery regimens between the assessed matches.

External load was monitored using global positioning system (GPS) devices (10-Hz, Catapult Sports, Melbourne, Australia). The devices were fitted to the upper back of each player using an elastic harness (Catapult Sports, Melbourne, Australia). The reliability and accuracy of 10 Hz GPS devices have been reported previously [41]. The studied variables comprise the following: total distance covered (TDC), distance covered while running at high speeds (DC 18.0–20.9 km·h^−1^), distance covered while running at very high speeds (DC 21.0–23.9 km·h^−1^), distance covered while sprinting (DC > 24.0 km·h^−1^), numbers of high accelerations (>2.5 m·s^−2^), numbers of high decelerations (<2.5 m·s^−2^) and peak speed (km·h^−1^). All the variables analysed were expressed in relative values per minute. Variables were classified in accordance with previous studies [42,43].

### 2.3. Statistics

Data are presented as mean ± standard deviation (SD). All variables presented a normal distribution (Shapiro-Wilk Test). A one-way analysis of variance (ANOVA) was used to determine differences between teams and playing positions. In the event of a significant difference, Bonferroni’s post hoc tests were used to identify any localized effects. Differences between groups and positions were analysed for practical significance using magnitude-based inferences by pre-specifying a 0.2 between-subject SD as the smallest worthwhile effect [44]. The standardized difference or effect size (ES, 90% confidence limit [90%CL]) in the selected variables was calculated. Threshold values for assessing the magnitudes of the ES (changes as a fraction or multiple of baseline standard deviation) were <0.20, 0.20, 0.60, 1.2 and 2.0 for trivial, small, moderate, large and very large, respectively [44].

## 3. Results

Comparisons of the external loads during the three matches in congested fixtures are shown in Table 1 and Figure 1. The statistical analysis showed significant differences in the numbers of high decelerations between M1 and M3 (*p* < 0.05).

The running activity of each position is shown in Table 2, Figure 2 and Figure 3. The analysis indicated that in at least one variable of DC 18.0–20.9 km·h^−1^, 21.0–23.9 km·h^−1^ and >24.0 km·h^−1^ all positions showed significant differences in the three matches during the congested period (*p* < 0.05). The MF achieved a significantly higher DC 18.0–20.9 km·h^−1^ in M3 than in the other matches (MF:M1 vs. M3, *p* = 0.04, ES = 0.12; MF:M2 vs. M3, *p* = 0.02, ES = 0.16), and the ST achieved a significantly higher TDC in M2 than in M3 (ST:M2 vs. M3, *p* = 0.03, ES = −0.39). The FB and CD achieved a significantly higher DC 21.0–23.9 km·h^−1^ in M1 than M3 and M2, respectively (FB:M1 vs. M3, *p* = 0.03, ES = −0.42; CD:M1 vs. M2, *p* = 0.01, ES = −0.52). Similarly, the MF achieved a significantly higher DC 21.0–23.9 km·h^−1^ in M3 than in M1 (MF:M1 vs. M3, *p* = 0.00, ES = 0.23). In M3, a significantly lower DC > 24.0 km·h^−1^ was found than in M1 for the CD (CD:M1 vs. M3, *p* = 0.01, ES = −0.35) and ST (ST:M1 vs. M3, *p* = 0.00, ES = −0.08) positions, but these were higher than in M2 for the MF (MF:M2 vs. M3, *p* = 0.00, ES = 0.23). The WM showed significantly more DC > 24.0 km·h^−1^ in M2 and M3 than in M1 (WM:M1 vs. M2, *p* = 0.00, ES = 0.89; WM:M1 vs. M3, *p* = 0.00, ES:1.03p). The WM and MF exhibited significantly more high decelerations in M3 than in M1 (WM:M1 vs. M3, 0.00, ES = 01.85; MF:M1 vs. M3, *p* = 0.00, ES = 0.56). The CD (CD:M2 vs. M3, *p* = 0.03, ES = 0.07), ST (ST:M2 vs. M3, *p* = 0.03, ES = 0.47) and MF (MF:M2 vs. M3, *p* = 0.00, ES = 0.35) also showed more high decelerations in M3 than in M2.

## 4. Discussion

This study aimed to compare the external load of professional youth soccer players in matches played in a congested period and evaluate the positional running requirements of each match. This is the first research undertaken during the Regular Championship to assess the running performance of Under 14, Under 16 and Under 19 elite soccer players in a congested schedule. Our results showed that when playing three matches in the same week, youth soccer players did not show differences in their running performance except in the number of decelerations between M1 and M3. There were differences in the running requirements based on their playing position, showing important variability in their behaviour during a congested period.

The soccer players involved in our intervention did not display differences in total TDC, DC 18–20.9 km·h^−1^, DC 21.0–23.9 kh·h^−1^, DC > 24.0 km·h^−1^, numbers of high accelerations >2.5 m·s^−2^ and peak speed between the matches in a congested calendar. An important issue that could explain the similarity of our studied variables is that when soccer players have to participate in three matches during the same week, they may consciously adopt a pacing strategy to maintain high-intensity actions [8,45], achieving relatively uniform performances between matches. Although most of the studied variables showed no significant change, differences in the number of high decelerations between M1 and M3 were found. High decelerations have been highlighted as an important variable for understanding the physical demands on team sport players [46], but currently, despite the great interest in deceleration demands in soccer, there is a lack of a practical and concise approach to scientific research [47]. Taking into account that deceleration loads provoke high mechanical and metabolic demands [48,49] and are related to post-match muscle damage and fatigue [50], soccer coaches should pay attention to youth players’ recovery during a congested period, especially as the number of matches increases. Unfortunately, the contextual variables of the analysed matches were not studied. Considering that context can influence match outcomes [51,52,53], these aspects may have influenced the players’ behaviour.

It is difficult to compare the results obtained in this study with the previously published literature because there is a controversial and limited amount of available research in this area. While some authors have affirmed that soccer players showed increased running performances between the matches of a congested period [25], others have indicated that running activity was reduced [18]. Differences between our data and the previous studies could be because the sample in our study comprised Under 14, Under 16 and Under 19 professional youth players, while other studies monitored professional adult soccer players. The literature has demonstrated that youth and professional soccer players have different running behaviours when their match performances are assessed, affirming that data derived from a given population may not be relevant to other populations [54].

This investigation emphasized the positional running requirements developed in a congested period, revealing differences in all positions between the matches for some of the studied variables. Our data are in line with the published literature, finding that all positions showed differences in some parameters related to high-speed running (DC 18.0–20.9 km·h^−1^, 21.0–23.9 km·h^−1^ or >24.0 km·h^−1^). These findings are also in line with a recent investigation that analysed the inter-position diversity between match changes in measures of match physical performance, which demonstrated a higher match-to-match variation in distance covered using high-intensity running (≥18 km·h^−1^) [29,31]. Similar findings have been found in highly trained youth soccer players, where the between-match variability in high-intensity and very high-intensity activities was substantially higher than for the total distance covered [30]. A recent systematic review of match demands in youth soccer has demonstrated that running requirements demands were specific to the player’s position [54]. In the same way, our results reinforce the idea of specific positional running requirements during congested periods [55,56,57]. These results highlight the importance of contextualizing matches and attending to variables such as formations, score lines or home-away locations because they affect running performance [8].

Although the current investigation adds insightful information about the external load during a congested period in soccer, some limitations must be considered. A small sample of players was included in the analysis. The need to be on the field for at least 75% of the total time in three matches together with the use of players from a professional academy limited the accuracy of the sample. Secondly, this study has been carried out with fixed speed thresholds. Even though these thresholds provide useful information regarding player development and allow direct outcome comparisons between studies [54], further investigations should check what happens when individualizing speed thresholds are used to provide a more accurate representation of match running loads in youth soccer players. Another important point to note is that contextual factors such as the standard of the opponents, scores, period during the season and classification were not considered. Given that these aspects influence the running behaviour of soccer players, future studies should be performed, adding these elements. Lastly, this study did not measure any internal load variables that could minimize player fatigue. In physiological terms, the possible effects of genetics on athletic performance could be taken into account [58]. Future studies should try to evaluate the internal load and effects of genetics by increasing the sample during congested periods.

## 5. Conclusions

This study found differences only in the number of high decelerations in matches in congested periods. Coaches should therefore pay special attention to the fatigue produced by this variable. Further investigations should further assess whether decelerations are decisive for the overall performance of the team during periods of maximum intensity such as congested periods. Secondly, significant differences in running requirements have been found between matches according to player position. Consequently, individualized training protocols should be used when professional youth soccer players experience a congested period. Thus, the scientific evidence revealed by this investigation responds to the complexity of interactions established between extrinsic and intrinsic factors relating to players and teams, thereby providing tools to ensure efficient training through specific stimuli [59].

## Figures and Tables

**Figure 1 jfmk-07-00108-f001:**
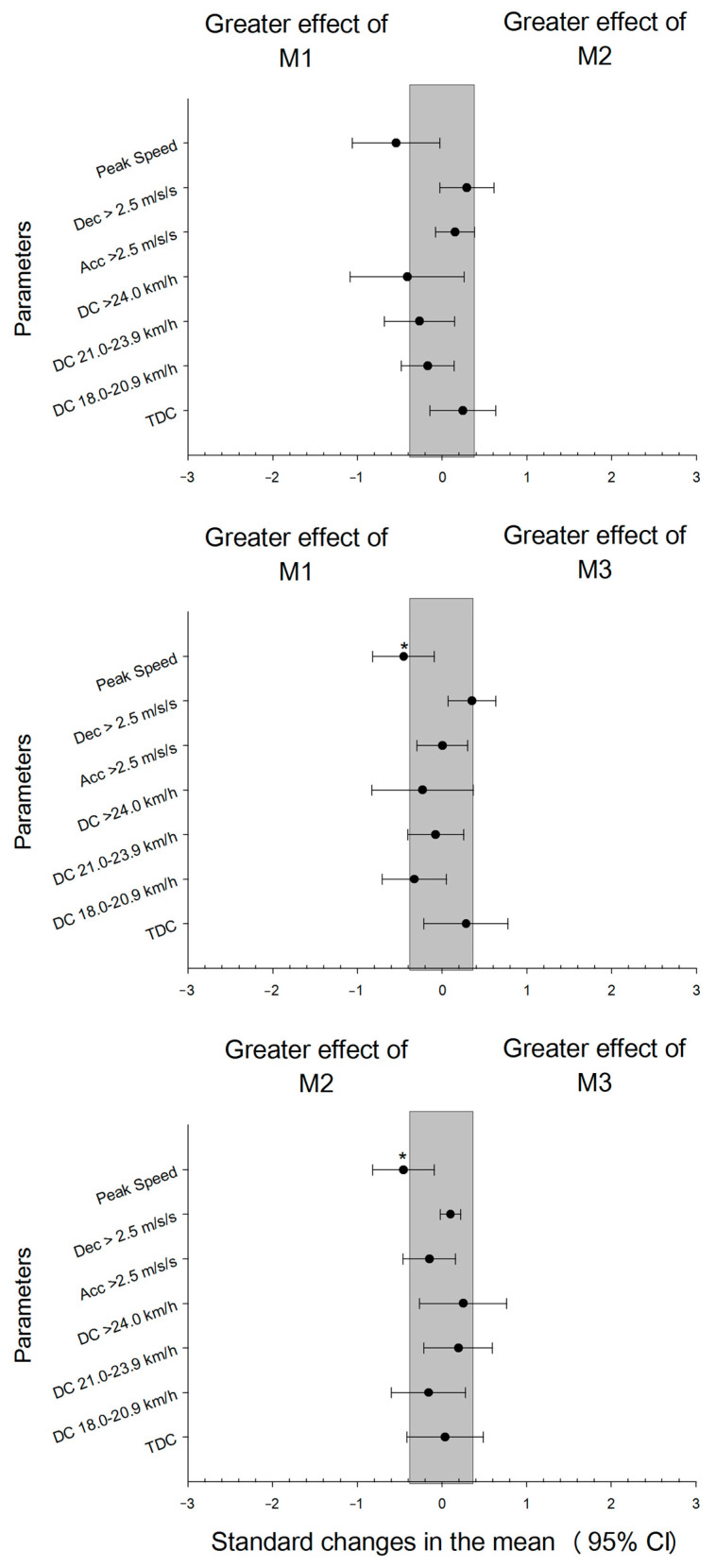
Comparison of external load during three matches in a congested fixture. * *p* < 0.05.

**Figure 2 jfmk-07-00108-f002:**
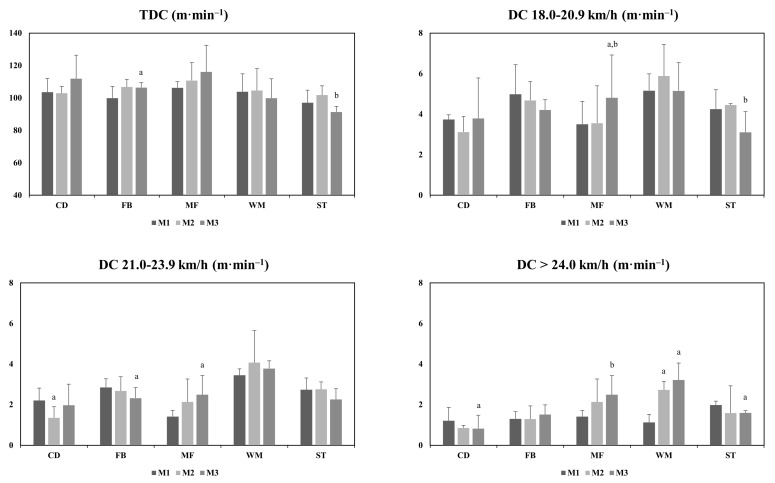
Comparison of distances covered at different speed thresholds during the three matches in a congested fixture. TDC = Total distance covered; CD = Central Defenders; FB = Fullbacks; MF = Midfielders; WM = Wide Midfielders; ST = Strikers; a = significant difference with M1; b = significant difference with M2.

**Figure 3 jfmk-07-00108-f003:**
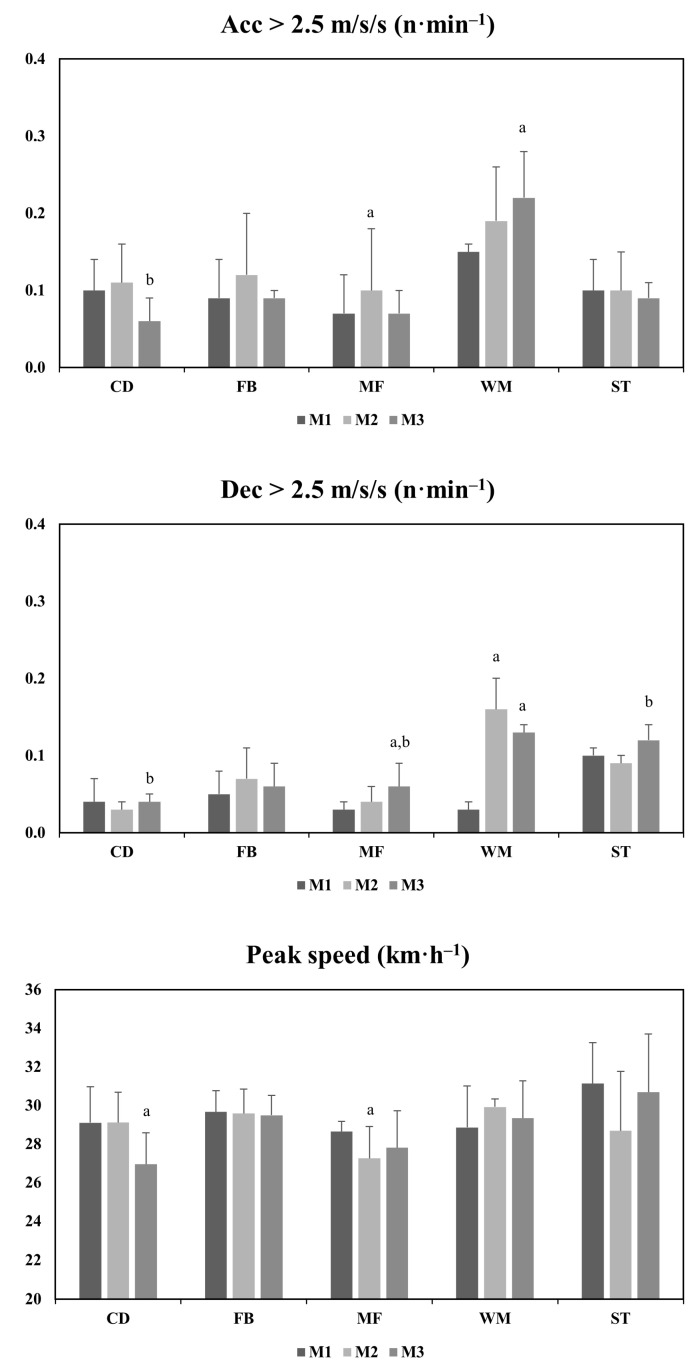
Comparison of accelerations, decelerations and peak speed during the three matches in a congested fixture. Acc = Accelerations; Dec = Decelerations; CD = Central Defenders; FB = Fullbacks; MF = Midfielders; WM = Wide Midfielders; ST = Strikers; a = significant difference with M1; b = significant difference with M2.

**Table 1 jfmk-07-00108-t001:** External load during the three matches in a congested fixture.

	Values		*p* Value	%	Q
M1 vs. M2					
TDC (m·min^−1^)	104.4 ± 8.1	106.1 ± 7.9	0.28	58/39/3	Possibly, may (not)
DC 18.0–20.9 km·h^−1^ (m·min^−1^)	4.6 ± 1.5	4.5 ± 1.6	0.36	44/53/3	Possibly, may (not)
DC 21.0–23.9 km·h^−1^ (m·min^−1^)	2.6 ± 1.0	2.5 ± 1.2	0.27	61/35/3	Possibly, may (not)
DC > 24.0 km·h^−1^ (m·min^−1^)	1.4 ± 0.6	1.3 ± 0.9	0.30	70/23/7	Possibly, may (not)
Acc > 2.5 m·s^−2^ (n·min^−1^)	0.10 ± 0.05	0.12 ± 0.07	0.26	36/63/1	Possibly, may (not)
Dec > 2.5 m·s^−2^ (n·min^−1^)	0.05 ± 0.03	0.06 ± 0.05	0.14	68/31/1	Possibly, may (not)
Peak speed (km·h^−1^)	29.4 ± 1.5	28.6 ± 2.2	0.08	87/12/1	Likely, probable
M1 vs. M3					
TDC (m·min^−1^)	104.4 ± 8.1	106.7 ± 12.3	0.34	61/34/6	Possibly, may (not)
DC 18.0–20.9 km·h^−1^ (m·min^−1^)	4.6 ± 1.5	4.3 ± 1.7	0.14	72/27/1	Possibly, may (not)
DC 21.0–23.9 km·h^−1^ (m·min^−1^)	2.6 ± 1.0	2.6 ± 0.9	0.69	26/66/8	Possibly, may (not)
DC > 24.0 km·h^−1^ (m·min^−1^)	1.4 ± 0.6	1.5 ± 1.0	0.50	54/35/11	Possibly, may (not)
Acc > 2.5 m·s^−2^ (n·min^−1^)	0.10 ± 0.05	0.10 ± 0.06	0.99	13/74/13	Unlikely, probable
Dec > 2.5 m·s^−2^ (n·min^−1^)	0.05 ± 0.03	0.07 ± 0.04	0.04	82/18/0	Likely, probable
Peak speed (km·h^−1^)	29.4 ± 1.5	28.7 ± 2.0	0.62	88/12/0	Likely, probable
M2 vs. M3					
TDC (m·min^−1^)	106.1 ± 7.9	106.7 ± 12.3	0.88	27/55/19	Possibly, may (not)
DC 18.0–20.9 km·h^−1^ (m·min^−1^)	4.5 ± 1.6	4.3 ± 1.7	0.53	44/48/8	Possibly, may (not)
DC 21.0–23.9 km·h^−1^ (m·min^−1^)	2.5 ± 1.2	2.6 ± 0.9	0.42	49/46/5	Possibly, may (not)
DC > 24.0 km·h^−1^ (m·min^−1^)	1.3 ± 0.9	1.5 ± 1.0	0.41	58/11/31	Possibly, may (not)
Acc > 2.5 m·s^−2^ (n·min^−1^)	0.12 ± 0.07	0.10 ± 0.06	0.39	3/57/40	Possibly, may (not)
Dec > 2.5 m·s^−2^ (n·min^−1^)	0.06 ± 0.05	0.07 ± 0.04	0.17	8/92/0	Unlikely, probable
Peak speed (km·h^−1^)	28.6 ± 2.2	28.7 ± 2.0	0.81	14/79/6	Possibly, may (not)

Data are presented as mean ± SD; % = percentage of change; Q = qualitative value; TDC = Total distance covered; Acc = Accelerations; Dec = Decelerations.

**Table 2 jfmk-07-00108-t002:** External load during the three matches in a congested fixture according to the playing position.

		TDC (m·min^−1^)	DC 18.0–20.9 km·h^−1^ (m·min^−1^)	DC 21.0–23.9 km·h^−1^ (m·min^−1^)	DC > 24.0 km·h^−1^ (m·min^−1^)	Acc > 2.5 m·s^−2^ (n·min^−1^)	Dec > 2.5 m·s^−2^ (n·min^−1^)	Peak Speed (km·h^−1^)
CD	M1	103.5 ± 8.4	3.7 ± 0.2	2.2 ± 0.6	1.2 ± 0.7	0.10 ± 0.04	0.04 ± 0.03	29.1 ± 1.9
	M2	103.0 ± 4.1	3.1 ± 0.8	1.4 ± 0.5	0.9 ± 0.1	0.11 ± 0.05	0.03 ± 0.01	29.1 ± 1.6
	M3	111.9 ± 14.4	3.8 ± 2.0	2.0 ± 1.0	0.8 ± 0.7	0.06 ± 0.03	0.04 ± 0.01	27.0 ± 1.6
FB	M1	99.9 ± 7.2	5.0 ± 1.5	2.9 ± 0.4	1.3 ± 0.4	0.09 ± 0.05	0.05 ± 0.03	29.7 ± 1.1
	M2	106.8 ± 4.7	4.7 ± 0.9	2.7 ± 0.7	1.3 ± 0.7	0.12 ± 0.08	0.07 ± 0.04	29.6 ± 1.3
	M3	106.4 ± 3.0	4.2 ± 0.5	2.3 ± 0.5	1.5 ± 0.5	0.09 ± 0.01	0.06 ± 0.03	29.5 ± 1.0
MF	M1	106.3 ± 3.8	3.5 ± 1.1	1.4 ± 0.3	0.9 ± 0.3	0.07 ± 0.05	0.03 ± 0.01	28.6 ± 0.6
	M2	110.8 ± 11.0	3.6 ± 1.8	2.1 ± 1.1	0.8 ± 0.5	0.10 ± 0.08	0.04 ± 0.02	27.3 ± 1.7
	M3	116.0 ± 16.5	4.8 ± 2.1	2.5 ± 0.9	1.2 ± 0.9	0.07 ± 0.03	0.06 ± 0.03	27.8 ± 1.9
WM	M1	103.9 ± 11.0	5.2 ± 0.8	3.5 ± 0.3	1.1 ± 0.4	0.15 ± 0.01	0.03 ± 0.01	28.9 ± 2.1
	M2	104.6 ± 13.5	5.9 ± 1.6	4.1 ± 1.6	2.7 ± 0.4	0.19 ± 0.07	0.16 ± 0.04	29.9 ± 0.4
	M3	99.9 ± 12.0	5.2 ± 1.4	3.8 ± 0.4	3.2 ± 0.8	0.22 ± 0.06	0.13 ± 0.01	29.3 ± 1.9
ST	M1	97.1 ± 7.8	4.3 ± 1.0	2.7 ± 0.6	2.0 ± 0.2	0.10 ± 0.04	0.10 ± 0.01	31.1 ± 2.1
	M2	101.8 ± 5.7	4.5 ± 0.1	2.8 ± 0.4	1.6 ± 1.3	0.10 ± 0.05	0.09 ± 0.01	28.7 ± 3.1
	M3	91.4 ± 3.5	3.1 ± 1.0	2.3 ± 0.5	1.6 ± 0.1	0.09 ± 0.02	0.12 ± 0.02	30.7 ± 3.0

Data are presented as mean ± SD. TDC = Total distance covered; Acc = Accelerations; Dec = Decelerations; CD = Central Defenders; FB = Fullbacks; MF = Midfielders; WM = Wide Midfielders; ST = Strikers.

## Data Availability

Not applicable.

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
