# Peer review of "Congested Period in Professional Youth Soccer Players Showed a Different High Decelerations Profile in the Group Performance and a Specific Positional Behaviour"

_jfmk, 2022, doi:10.3390/jfmk7040108_

Round 1
Reviewer 1 Report
Dear Author,
Thank you for your effort. The manuscript is well written. As explained below, some points need to be addressed before publication.
- The authors do not provide a power calculation. Was a sample size calculation undertaken? What is the power of the study with twenty subjects?
-Why were effect sizes not used? It would be better if the effect size was also added.
Reviewer 2 Report
The manuscript is well written and the results are clearly described.
The authors might mention the possible effect of genetic on athletic performance. This could open avenues for further study. (246)
https://dergipark.org.tr/en/download/article-file/2042367
https://www.researchgate.net/publication/344220767_Kosucularda_ACTN3_ve_ACE_Genlerinin_Sportif_Performansa_Etkisi
Reviewer 3 Report
Comments to Authors:
This manuscript provides insight into the demands of the congested competitive calendar in youth soccer athletes which I feel is wonderful and does adds a unique data set. I have a few minor concerns and questions that I would like to see addressed before providing a recommendation for publication. While there are many citations provided within this manuscript there are a few that I am a bit surprised that were omitted as they provide evidence directly related to sprinting in youth soccer populations during match
Introduction:
Overall, the introduction provides a sufficient foundation for this study. I feel that there are a few points that may warrant further detail to provide more information to the reader and can be added into the discussion later.
Line 54-56: Is the no statistical difference in distance covered in reference to the congested period or between age groups? As you pool the age groups in the analysis this stood out as a point that could use some clarification.
As demands between matches can vary when it comes to some of the metrics used in this study, I think showing some of this variation in the introduction would be beneficial as it can help explain the differences seen between studies.
Methods:
There is evidence that high-speed running thresholds do not translate across, as there are differences in sprint time at given yardages (Mendez-Villanueva et al 2011, Journal of Sport Science 29(5): 477-484, Mainer-Pardos et al 2001 BMC Sports Science, Med, Rehab 13) and evidence that based on age there are differences in the amount of distance covered in each of these speed zones (Gato and Saward 2020, JSCR 34(6):1564-1573). This would appear to have implications to this study in terms of the positional comparisons, as the sample size for each position is uneven with different a different number of subjects from each roster. A more through explanation in why these thresholds were used should be placed either in the methods or the discussion should address this.
Discussion:
In contrast to the point raised in the methods a recent systematic review of match demands in youth soccer (Palucci et al 2019, Sports Medicine 49 289-318) provides some evidence to support that your findings match those in the literature when it comes to the positional differences and would provide excellent support for this manuscript from the perspective of the trends that you see are consistent in the literature.
Again, similar to the discussion I feel that there is room to add depth to the discussion as the data presented does provide answers to some questions and triggers more questions that may be worth exploring.
Figures:
All the figures need to have the units add to the vertical axis.
Tables:
All the tables need to have the units added to each variable. While I can assume that it is meters, I should not have to make such an assumption. If this is the case then travelling roughly 100 meters a match in Total distance is rather low as previous finding are upwards of 10000 m range.
Also, if the acceleration and deceleration values are in the number of occurrences as mentioned in the methods, how can they be 0.05 as we see in the M1 vs M2. If that is the case then I think there is a larger issue at hand.
